# Valence and salience encoding in the central amygdala

**Mi-Seon Kong[1], Ethan Ancell[2], Daniela M Witten[2,3], Larry S Zweifel[1,4]***

[1]Department of Psychiatry and Behavioral Sciences, University of Washington, Seattle, United States; [2]Department of Statistics, University of Washington, Seattle, United States; [3]Department of Biostatistics, University of Washington, Seattle, United States; [4]Department of Pharmacology, University of Washington, Seattle, United States

## eLife Assessment

This **useful** work reveals differential activity to food and shock outcomes in central amygdala GABAergic neurons. Evidence supports claims of unconditioned stimulus activity that changes with learning. **Compelling** evidence that the circular shift method rigorously identifies functional neuron types is also presented. However, the evidence regarding claims related to valence or salience signaling in these neurons is **incomplete**. This work will be of interest to neuroscientists studying sensory processing and learning in the amygdala.

***For correspondence:**
larryz@uw.edu

**Competing interest:** The authors declare that no competing interests exist.

**Abstract** The central amygdala (CeA) has emerged as an important brain region for regulating both negative (fear and anxiety) and positive (reward) affective behaviors. The CeA has been proposed to encode affective information in the form of valence (whether the stimulus is good or bad) or salience (how significant is the stimulus), but the extent to which these two types of stimulus representation occur in the CeA is not known. Here, we used single cell calcium imaging in mice during appetitive and aversive conditioning and found that majority of CeA neurons (~65%) encode the valence of the unconditioned stimulus (US) with a smaller subset of cells (~15%) encoding the salience of the US. Valence and salience encoding of the conditioned stimulus (CS) was also observed, albeit to a lesser extent. These findings show that the CeA is a site of convergence for encoding oppositely valenced US information.

## Introduction

The CeA is comprised of multiple genetically distinct cell types that contribute to both appetitive and aversive processes (*Cai et al., 2014*; *Douglass et al., 2017*; *Fadok et al., 2018*; *Hardaway et al., 2019*; *Haubensak et al., 2010*; *Isosaka et al., 2015*; *Kim et al., 2017*; *Li et al., 2013*; *McCullough et al., 2018*; *Warlow et al., 2020*; *Yu et al., 2017*). Subsets of cells in the CeA encode conditioned and/or unconditioned fear stimulus information (*Ciocchi et al., 2010*; *Duvarci et al., 2011*; *Fadok et al., 2017*; *Li et al., 2013*; *Sanford et al., 2017*; *Yang et al., 2023*; *Yu et al., 2017*), as well as appetitive information (*Douglass et al., 2017*; *Hardaway et al., 2019*; *Yang et al., 2023*). Plasticity within neurons of the CeA (*Fu and Shinnick-Gallagher, 2005*) contributes to both fear-related learning (*Li et al., 2013*; *Penzo et al., 2014*; *Sanford et al., 2017*) and reward-associated learning (*Yang et al., 2023*). Cells implicated in fear processing have also been shown to regulate consummatory behaviors (*Cai et al., 2014*; *Douglass et al., 2017*; *Yang et al., 2023*; *Yu et al., 2017*), suggesting that subsets of CeA neurons encode either incentive salience (the significance of the stimulus regardless of valence) or valence (whether it is a positive or negative outcome).

**Figure 1.** Acquisition of calcium signals during Pavlovian appetitive and fear conditioning. (**A**) CeA-GABAergic neurons were labelled with GCaMP6m, and their activity was recorded using a miniature microscope (Inscopix) via an implanted GRIN lens (left). A representative image of GCaMP6m expression and lens placement is shown on the right. Scale bar, 0.5 mm. (**B**) Behavioral paradigm of the baseline, appetitive, and fear conditioning. To counterbalance the order of the two valence conditioning paradigms, there were two groups: appetitive → fear (5 mice) and fear → appetitive (5 mice) after the baseline session. (**C**) Time spent near the food hopper during CS$^{Food}$ (left, day: $F=7.422$, $P=0.0005$), latency to procure a pellet (middle, day: $F=20.41$, $P<0.0001$), and success rate (right, day: $F=14.69$, $P=0.0017$) over 10 days of appetitive learning (mean ± s.e.m., grey lines represent individual data, n=10). (**D**) Freezing behavior during CS$^{Shock}$ (trial: $F=13.49$, $P<0.0001$) throughout 10 trials of fear conditioning (mean ± s.e.m., grey lines represent individual data, n=10). Detailed information about statistical results is provided in **Supplementary file 1**.

The online version of this article includes the following figure supplement(s) for figure 1:

**Figure supplement 1.** GRN lens placements.

Specific subsets of neurons within the CeA have been shown to contribute to multiple aspects of appetitive and aversive behaviors (**Balleine and Killcross, 2006**; **Kong and Zweifel, 2021**; **Pignatelli and Beyeler, 2019**) and it was recently shown that subpopulations of CeA neurons provide scalar signals to oppositely valenced stimuli to promote learning (**Yang et al., 2023**). However, the full extent to which the CeA encodes the salience or valence of positive and negative affective information remains unresolved.

To address the extent of valence and salience encoding within the CeA, we used single cell calcium imaging to longitudinally track neurons during Pavlovian reward and Pavlovian fear conditioning. We found that the majority of CeA neurons are responsive to either the reward or the fear US. Although a subset of cells displays salience-like encoding of the US, most CeA neurons were selectively responsive to the valence of the US. We observed six distinct types of valence encoding that was either selective for one or the other USs or oppositely encoded US information. Salience and valence encoding was also observed in response to a reward- or fear-associated CS, but the magnitude of these responses and the number of cells responding to the CSs were

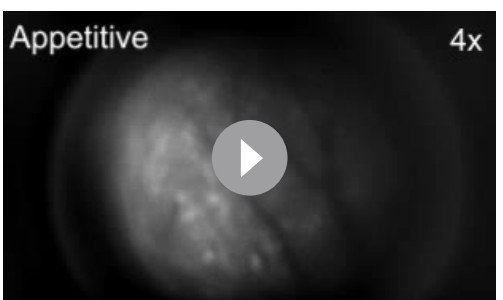

**Video 1.** CeA activity during the appetitive conditioning day 10.
https://elifesciences.org/articles/101980/figures#video1

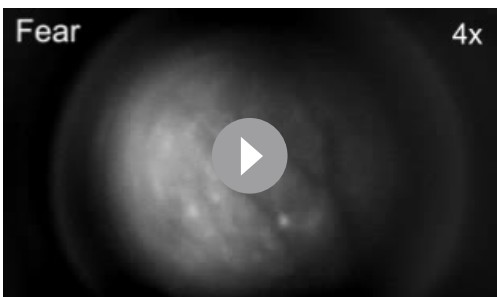

**Video 2.** CeA activity during the fear conditioning.
https://elifesciences.org/articles/101980/figures#video2

considerably smaller than the responses observed to the USs. These data indicate that the CeA encodes multiple aspects of positive and negative reinforcing stimuli, but the prevailing responses relate to the encoding of the valence of the US.

## Results

To monitor calcium dynamics in CeA neurons during positive and negative reinforcement, we selectively expressed GCaMP6m in GABAergic neurons of Vgat$^{IRES-Cre}$ (Slc32a1$^{tm2(cre)Lowl}$/J; JAX strain #: 016962) mice (4 males and 6 females) and implanted a gradient-index (GRIN) lens in the CeA (*Figure 1A*, *Figure 1—figure supplement 1*). Mice were conditioned in both Pavlovian appetitive and Pavlovian fear paradigms, with the conditioning order counterbalanced (*Figure 1B*, *Videos 1 and 2*). During appetitive learning, mice demonstrated a significant increase in time spent near the food hopper during the food-predicting CS (CS$^{Food}$), reduced latency to retrieve the food pellet, and an increased success rate in procuring the reward (*Figure 1C*). During fear conditioning, the animals learned that the shock-predicting CS (CS$^{Shock}$) was followed by an impending foot shock and displayed increased freezing behavior to the CS$^{Shock}$ as trials progressed (*Figure 1D*).

During conditioning, we observed responsive and non-responsive cells to both the appetitive and aversive US (*Figure 2A*). To resolve event-related calcium signals, we developed a variation of the 'circular shift' method[27] to stringently define cells with statistically significant responses (see Methods for detailed description). In short, we used circular shifting to generate artificial "null" calcium traces resembling a sample from a hypothetical population of neuron traces unrelated to the exact time of behavior events (Step 1, *Figure 2B*). Wilcoxon rank sum tests (comparing pre-stimulus vs. post-stimulus activity) were performed on recorded traces and "null" traces and summed across trials (Step 2 and Step 3, *Figure 2C and D*) to create a single test statistic for each neuron. P-values were obtained by comparing the summed Wilcoxon rank sum test statistics from actual data against the summed Wilcoxon rank sum test statistics from the artificially generated null distribution (Step 4, *Figure 2E*).

During Pavlovian appetitive conditioning, alignment to the cue presentation revealed no responsive cells to the CS$^{Food}$ or food delivery on day 1 (*Figure 3A*, top). However, on day 10, numerous cells were identified that were responsive to food delivery with either increases or decreases in calcium signals (*Figure 3A*, bottom). Analysis of responses aligned to food retrieval (head entry) revealed responsive cells on both day 1 and day 10 (*Figure 3B*). The number of cells showing increased calcium to head entry and the magnitude of the increase was larger on day 10 compared to day 1 (*Figure 3B–D*). Of the 872 cells recorded on day 10 of conditioning, a small proportion of cells were responsive to CS$^{Food}$ presentation, showing similar proportions of cells with increased or decreased calcium signals (*Figure 3E and F*). Of these CS$^{Food}$ responsive cells, the majority were responsive to both the CS and the food delivery (*Figure 3—figure supplement 1A and B*). In contrast to the CS$^{Food}$ responsive cells, a large number (64%) of CeA neurons were responsive to food delivery, with the largest proportion of cells showing excitatory responses (40%) compared to inhibited responses (24%) (*Figure 3G and H*).

Additional analysis of CeA neuron responses on day 10 of Pavlovian appetitive conditioning revealed that there was a trend toward decreased responsiveness to the CS$^{Food}$ late in conditioning compared to early in the excited neurons and a significant increase in the inhibitory response in late conditioning compared to early (*Figure 3—figure supplement 1C and D*). In response to food delivery, no differences were observed in food-excited neurons; however, a small but significant difference was observed in food-inhibited cells with an increase in the magnitude of the inhibited response late in conditioning compared to early (*Figure 3—figure supplement 1E and F*).

During Pavlovian fear conditioning, alignment of calcium signals to the CS$^{Shock}$ revealed that the greatest responses were to the US presentation (*Figure 4A*). Of the 519 cells recorded, a small percentage were either excited or inhibited by the CS$^{Shock}$ (*Figure 4B and C*). A large number of

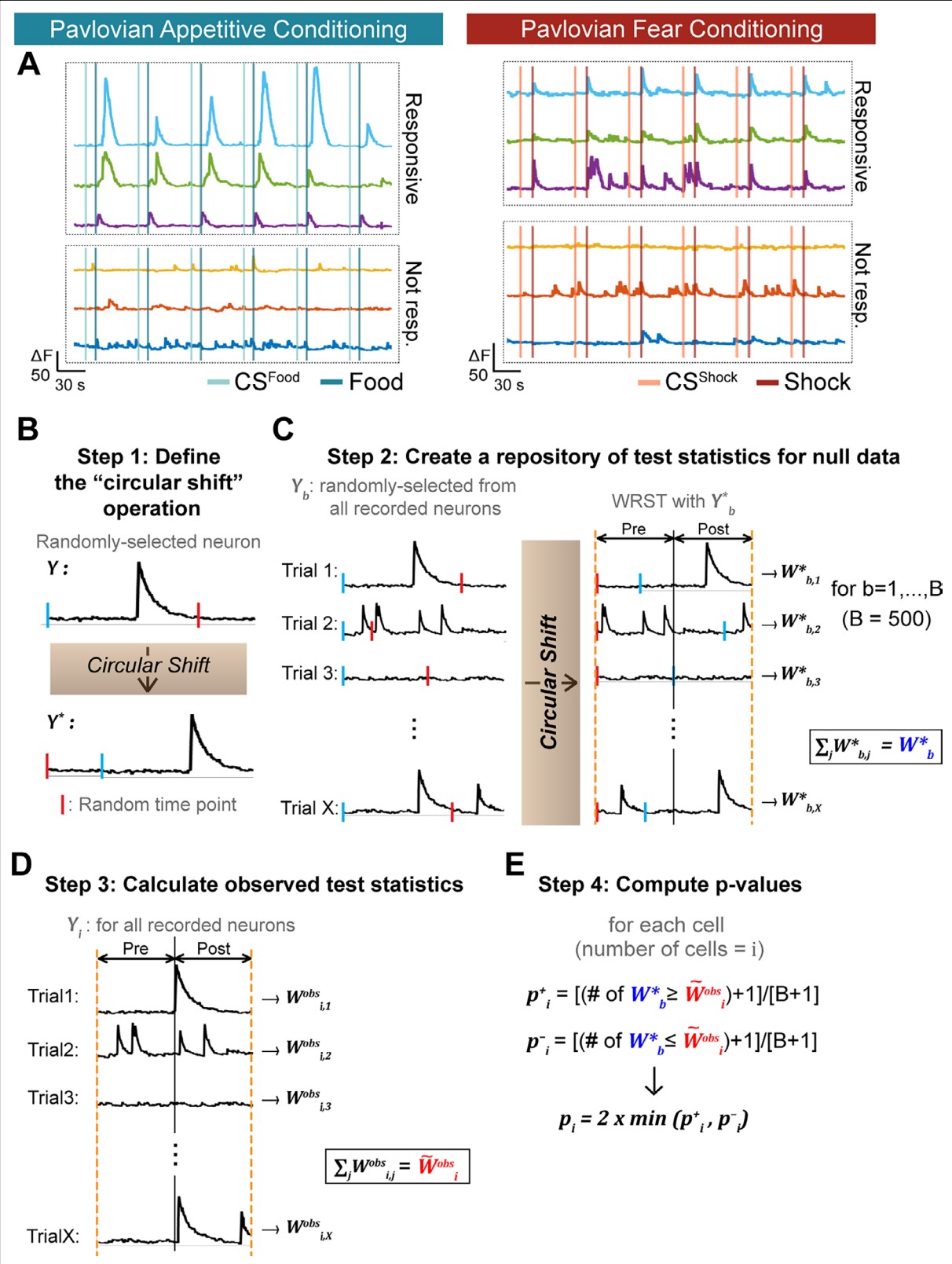

**Figure 2.** Determination of CeA responses to appetitive and fearful stimuli. (**A**) Left: Example traces of CeA neurons that were responsive (upper three traces) and not responsive to food reward (bottom three traces). Right: Example traces of CeA neurons that were responsive (upper three traces) and not responsive to foot shock (bottom three traces). (**B**) Step 1: Defining the "circular shift" operation. A neuron is randomly selected from the recorded neuron pool in our current study (denoted as $Y$). The calcium trace of the selected neuron is subjected to a circular shift operation, wherein it is moved

*Figure 2 continued on next page*

*Figure 2 continued*

from a randomly chosen time point (red bar). This operation results in a new calcium trace, as shown by the movement of the blue bar in (**B**); the traces that initially start with blue bars become red bar starting traces. (**C**) Step 2: Creating a repository of test statistics for null data. A neuron is selected at random; then, the activity in each of its trials (20 trials for appetitive or 10 trials for fear conditioning, X=the number of trials) undergoes 'circular shifting', where a random point (red bar) is chosen within the circular shifting range. Then, on each trial, the Wilcoxon rank sum test between pre-stimulus vs. post-stimulus periods is computed. The final test statistic $W_b^*$ representing a summary of all trials is obtained by summing the individual Wilcoxon rank sum test statistics over the trials with $W_b^* = \sum_j W_{b,j}^*$. Step 2 is repeated $B$ times (we used B=500), generating a null distribution of summed statistics $\{W_b^*\}_{b=1}^B$. (**D**) Step 3: calculating observed test statistics. Each observed neuron receives a summary test statistic $\widetilde{W}_i^{obs} = \sum_j W_{i,j}^{obs}$ obtained by summing the Wilcoxon rank sum test statistics from the individual trials. (**E**) Step 4: Computing p-values. A p-value for the i$^{th}$ neuron comes from a comparison of $\widetilde{W}_i^{obs}$ with $\{W_b^*\}_{b=1}^B$. For the i$^{th}$ neuron, $p_i^+$ (the number of $W_b^* \geq \widetilde{W}_i^{obs}$ +1 divided by B+1) and $p_i^-$ (the number of $W_b^* \leq \widetilde{W}_i^{obs}$ +1 divided by B+1) are calculated. The final p-value for the i$^{th}$ neuron (denoted as $p_i$) is 2 times the lesser of $p_i^+$ and $p_i^-$. A detailed description of this statistical analysis is provided in Methods.

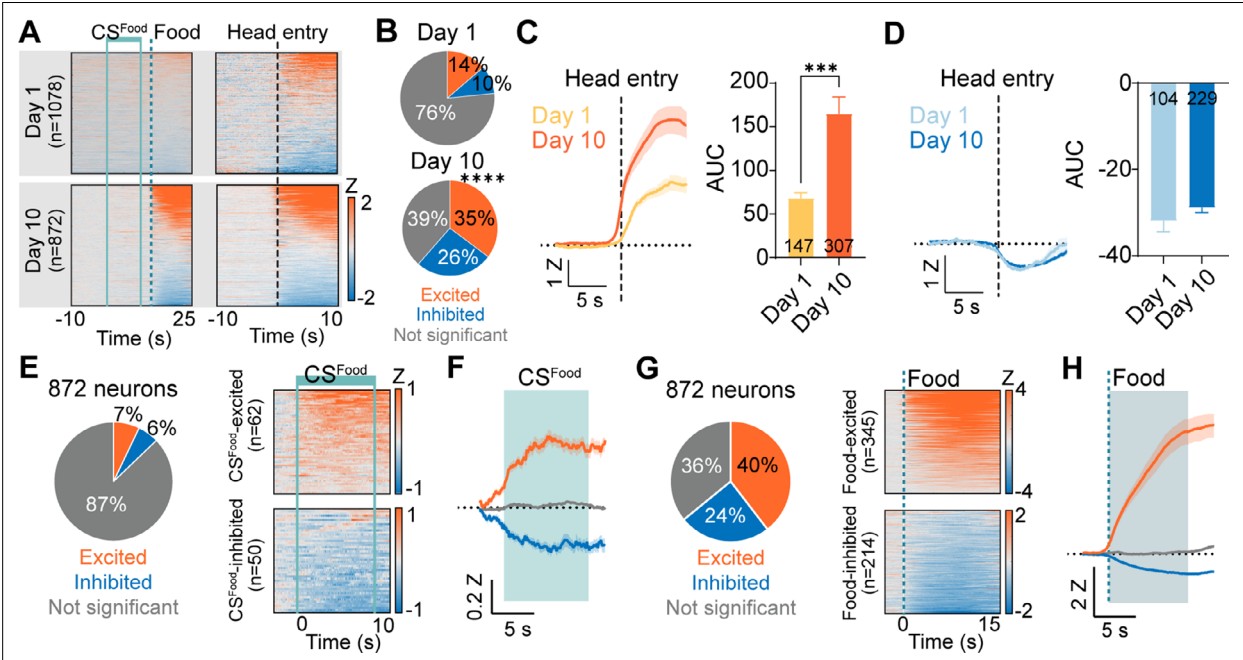

**Figure 3.** Responses of CeA neurons during Pavlovian appetitive conditioning. (**A**) Top: Heat maps of CeA neurons aligned to CS$^{Food}$ onset (top left, 1078 neurons) and head entry (top right) from day 1 of Pavlovian appetitive conditioning. Bottom: Heat maps of CeA neurons aligned to CS$^{Food}$ onset (bottom left, 872 neurons) and head entry (bottom right) from day 10 of Pavlovian appetitive conditioning. Solid mint lines indicate CS$^{Food}$, and a dotted darker mint line represents food delivery. Black dotted line represents the first head entry after food delivery. (**B**) Proportion of head entry responsive neurons between day 1 (top) and day 10 (bottom). (**C**) Left: The average Z-scored activity of head entry-excited neurons on day 1 (147 neurons) and day 10 (307 neurons) of appetitive conditioning during –10 s to 10 s after the first head entry of each reward delivery. The dark lines and shaded areas indicate the mean and standard error of the mean (s.e.m.). Right: Average area under the curve (AUC) for the Z-scored activity after head entry on day 1 compared to day 10 of appetitive conditioning (0–5 s after head entry, mean ± s.e.m.). (**D**) Left: The average Z-scored activity of head entry-inhibited neurons on day 1 (104 neurons) and day 10 (229 neurons) of appetitive conditioning during –10 s to 10 s after the first head entry of each reward delivery. Right: Average AUC for the Z-scored activity after head entry on day 1 compared to day 10 of appetitive conditioning (0–5 s after head entry). (**E**) Left: Proportion of CS$^{Food}$-excited (orange), CS$^{Food}$-inhibited (blue), and not significant neurons (grey) on day 10 of appetitive learning (total 872 neurons). Right: Heat maps of CS$^{Food}$-excited neurons (top, n=62) and CS$^{Food}$-inhibited neurons (bottom, n=50) from all trials (20 trials). Solid mint lines indicate CS$^{Food}$ onset and offset. (**F**) The average Z-scored activity of each response type (orange, CS$^{Food}$-excited; blue, CS$^{Food}$-inhibited; grey, not significant). The dark lines and shaded areas represent the mean and s.e.m. The mint area represents 10 s of CS$^{Food}$. (**G**) Left: Proportion of food-excited (orange), food-inhibited (blue), and not significant neurons (grey) on day 10 of appetitive learning (total 872 neurons). Right: Heat maps of food-excited neurons (top, n=345) and food-inhibited neurons (bottom, n=241) from all trials (20 trials). The dotted line indicates a food delivery (**H**) The average Z-scored activity of each response type (orange, food-excited; blue, food-inhibited; grey, not significant). The dark lines and shaded areas represent the mean and s.e.m. ***P<0.001, ****P<0.0001. Detailed information about statistical results is provided in *Supplementary file 1*.

The online version of this article includes the following figure supplement(s) for figure 3:

**Figure supplement 1.** CS$^{Food}$ and food response in the CeA neurons.

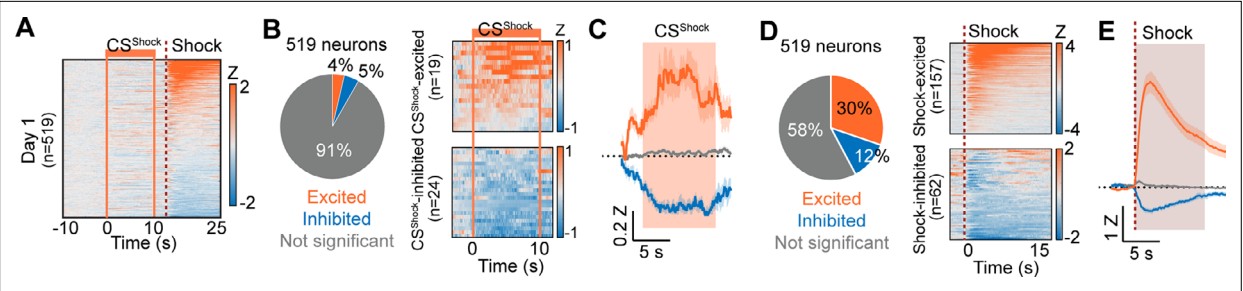

**Figure 4.** Responses of CeA neurons during Pavlovian fear conditioning. (**A**) Heat maps of CeA neurons aligned to $CS^{Shock}$ onset (519 neurons) from day 1 of Pavlovian fear conditioning. Solid orange lines indicate $CS^{Shock}$, and a dotted red line represents shock delivery. (**B**) Left: Proportion of $CS^{Shock}$-excited (orange), $CS^{Shock}$-inhibited (blue), and not significant neurons (grey) on day 1 of fear conditioning (total 519 neurons). Right: Heat maps of $CS^{Shock}$-excited neurons (top, n=19) and $CS^{Shock}$-inhibited neurons (bottom, n=24) from all trials (10 trials). Solid orange lines indicate $CS^{Shock}$ onset and offset. (**C**) The average Z-scored activity of each response type (orange, $CS^{Shock}$-excited; blue, $CS^{Shock}$-inhibited; grey, not significant). The dark lines and shaded areas represent the mean and s.e.m. The orange area represents 10 s of $CS^{Shock}$. (**D**) Left: Proportion of shock-excited (orange), shock-inhibited (blue), and not significant neurons (grey) on day 1 of fear conditioning (total 519 neurons). Right: Averaged heat maps of shock-excited neurons (top, n=157) and shock-inhibited neurons (bottom, n=62) from all trials (10 trials). A dotted red line represents shock delivery. (**E**) The average Z-scored activity of each response type (orange, shock-excited; blue, shock-inhibited; grey, not significant). The dark lines and shaded areas represent the mean and s.e.m.

The online version of this article includes the following figure supplement(s) for figure 4:

**Figure supplement 1.** $CS^{Shock}$ and shock response in the CeA neurons.

cells (42%) were responsive to the shock US with the largest proportion showing excitations (30%) compared to inhibitions (12%) (*Figure 4D and E*). Of the $CS^{Shock}$ responsive cells, an approximately equal number were responsive to the $CS^{Shock}$ only or the $CS^{Shock}$ and $US^{Shock}$ (*Figure 4—figure supplement 1A and B*). In contrast to the $CS^{Food}$ responsive cells during day 10 of Pavlovian appetitive conditioning, we did not observe any changes in the $CS^{Shock}$ responsive cells during early compared to late conditioning (*Figure 4—figure supplement 1C and D*). However, shock-excited CeA neurons showed a significant attenuation of the excited response in late conditioning compared to early conditioning (*Figure 4—figure supplement 1E*). No changes were observed in shock-inhibited cells (*Figure 4—figure supplement 1F*).

Qualitatively, food-excited and shock-excited neurons exhibited distinctive properties. Food-excited neurons showed more sustained activity in response to food, and their time to peak activity occurred significantly later than that of shock-excited neurons, likely reflecting the longer time needed to consume food compared to the passive delivery of the shock (*Figure 5—figure supplement 1A*). Additionally, the maximum Z-score of food-excited neurons was larger than that of shock-excited neurons, indicating that food was more salient for the CeA (*Figure 5—figure supplement 1B*).

To categorize cells as salience or valence encoding, we identified 303 neurons that appeared during both appetitive and fear conditioning (*Figure 5A*). Salience-encoding neurons (see Methods for detailed definition) were defined as those showing both excited or inhibited responses to food and shock, and comprised 15% of the total neurons (*Figure 5B and C*). In contrast, valence-encoding neurons were defined as those displaying differential activity to food and shock and represented 65% of the neurons (*Figure 5B–E*). Of these, 13.5% showed opposite responses to food or shock while the remaining 51% showed selective responses to one or the other USs (*Figure 5D and E*). The remaining 20% were categorized as non-responsive. These findings demonstrate that CeA neurons encode either the valence or the salience of the US, with the valence encoding being more prevalent.

To categorize salience- and valence-encoding neurons based on their responses to $CS^{Food}$ and $CS^{Shock}$, we analyzed the 303 registered neurons recorded during both reward and fear conditioning sessions (*Figure 1C and D*). Only 2% of neurons were classified as salience-encoding, while 19% were identified as valence-encoding neurons. Although representing a significantly smaller proportion of CS encoding neurons compared to the US encoding neurons (CS encoding neurons: salience = 7, valence = 57, not significant = 239; US encoding neurons: salience = 46, valence = 198, not significant = 59; $X^2$=215.4, $P$<0.0001, df = 2), these findings demonstrate that encoding of the valence of the CS is also the predominant response type of the CeA neurons.

It is possible that the low number of cells responsive to the $CS^{Shock}$ during conditioning reflects the identification of these cells during acquisition as opposed to fear memory retrieval. To address this,

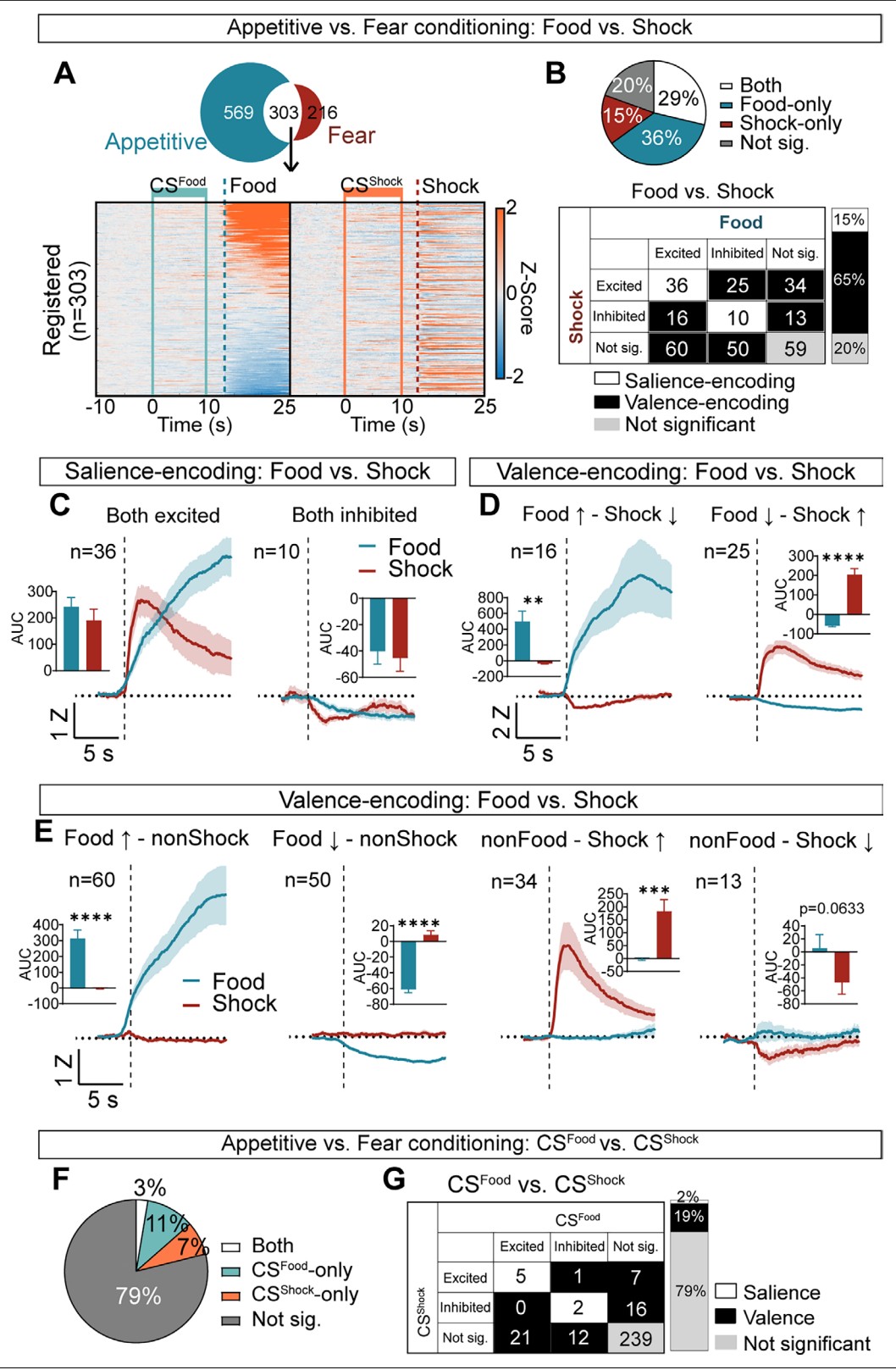

**Figure 5.** Salience and valence encoding in the CeA. (**A**) Among 872 neurons from appetitive conditioning and 519 neurons from fear conditioning, 303 neurons were registered during both learning. Heat maps of these 303 neurons during –10 s to 25 s after CS^Food onset from appetitive (left) and during –10 s to 25 s after CS^Shock onset from fear conditioning (right). All neurons are aligned to their activity to food. The same neuron is represented in the

*Figure 5 continued*

same row. Solid mint lines indicate CS[Food], and a dotted darker mint line represents food delivery, solid orange lines indicate CS[Shock], and a dotted red line represents shock delivery. (**B**) Top: the proportion of significant neurons to food and shock (both; white), food-only (blue), shock-only (red), and not significant to both (grey). Bottom: detailed response types of all possible categories (excited, inhibited, not significant) for food vs. shock. White represents salience-encoding neurons showing the same response types to food and shock (n=46, 15%), and black represents valence-encoding neurons showing different response types to food and shock (n=198, 65%). Grey represents not significant neurons (n=59, 20%). (**C**) Salience-encoding neurons: the average Z-scored activity of neurons that were excited to both food and shock (n=36, left) and inhibited to both food and shock (n=10, right). Inserted bar graphs indicate average AUC of the Z-scored activity after food and shock delivery (0–10 s, mean ± s.e.m.). (**D**) Valence-encoding neurons: the average Z-scored activity of neurons that were excited to food but inhibited to shock (n=16, left) and inhibited to food but excited to shock (n=25, right). Average AUC of the Z-scored activity after food and shock delivery are shown in inserted bar graphs (0–10 s, mean ± s.e.m.). (**E**) The average Z-scored activity of neurons that were excited to food but not responsive to shock (n=60, first), neurons that were inhibited to food but not responsive to shock (n=50, second), neurons that were not responsive to food but excited to shock (n=34, third), and neurons that were not responsive to food but inhibited to shock (n=13, fourth). Inserted bar graphs indicate average AUC of the Z-scored activity after food and shock delivery (0–10 s, mean ± s.e.m.). The dark lines and shaded areas represent the mean and s.e.m. (**F**) Proportion of significant neurons to CS[Food] and CS[Shock] (both; white), CS[Food]-only (mint), CS[Shock]-only (orange), and not significant to both (grey). (**G**) Detailed response type of all possible categories (excited, inhibited, not significant) for CS[Food] vs. CS[Shock]. White represents salience-encoding neurons showing the same response types to CS[Food] and CS[Shock] (n=7, 2%), and black represents valence-encoding neurons showing different response types to CS[Food] and CS[Shock] (n=57, 19%). Grey represents not significant neurons (n=239, 79%). *$P < 0.05$, ****$P < 0.0001$. Detailed information about statistical results is provided in *Supplementary file 1*.

The online version of this article includes the following figure supplement(s) for figure 5:

**Figure supplement 1.** Characteristics of CS and US responses in the CeA.

**Figure supplement 2.** Determining responsive neurons using the "theoretical" distribution of the Wilcoxon rank sum test, which assumes independence between timepoints.

**Figure supplement 3.** The order of valence conditioning did not affect the encoding of food, shock, and CSs in the CeA.

we analyzed CeA neuron responses to the CS[Food] and CS[Shock] twenty-four hours following fear conditioning (post-test) and compared these responses to a baseline pre-conditioning session. The number of responsive cells during the post-test was less than those observed during conditioning and did not differ from the proportion observed prior to conditioning (*Figure 5—figure supplement 1E–J*). Another possible reason for the fewer than expected CS[Food] and CS[Shock] responsive cells is the analysis we used to classify these neurons. To address this, we used a more conventional Wilcoxon sign rank test on each trial relative to the pre-CS period combined with distribution cutoff (*Figure 5—figure supplement 2A–C*). The number of CeA neurons excited or inhibited by the food US or shock US was similar to what was identified using the modified wrapping method (*Figure 5—figure supplement 2D–G*). Likewise, the number of cells responsive to the CS[Food] was similar (*Figure 5—figure supplement 2J and K*). CeA neurons excited or inhibited by the CS[Shock] were more prevalent using the sign rank method compared to the wrapping method (10% excited and 13% inhibited vs. 4% excited and 5% inhibited; *Figure 5—figure supplement 2L and M*). These marginal differences indicate that the classification method is not a major contributing factor to the observed results and confirm that CeA neurons as a whole are more responsive to the US than to the CS.

We next asked whether the order of valence conditioning (appetitive → fear or fear → appetitive) had any effect on the encoding of subsequent valence conditioning. Among the 303 registered neurons, 165 neurons belonged to the appetitive → fear group (*Figure 5—figure supplement 3A*), and 138 neurons were from the fear → appetitive group (*Figure 5—figure supplement 3E*). Our analysis revealed the following key findings: (i) CeA neurons were more responsive to food compared to shock regardless of the conditioning order (*Figure 5—figure supplement 3B and F*), (ii) the ratios of salience-, valence-encoding, and not significant neurons between the two groups were not significantly different (appetitive → fear group US: salience = 22, valence = 115, not significant = 28 vs. fear → appetitive group US: salience = 24, valence = 83, not significant = 31, $X2^2$ = 3.029, p=0.2199, df = 2, *Figure 5—figure supplement 3B and F*; appetitive → fear group CS: salience = 2, valence = 33, not

significant = 109 vs. fear → appetitive group CS: salience = 4, valence = 35, not significant = 120, $X2^2$ = 0.5126, p=0.7739, df = 2, *Figure 5—figure supplement 3C and G*), and (iii) the conditioning order did not preferentially recruit CeA neurons based on their topographical locations (*Figure 5—figure supplement 3D and H*), indicating that neurons encoding salience or valence were intermingled.

Finally, we investigated whether the responses to the USs or CSs were qualitatively biased by the initial valence conditioning experience. We found that the response to food before shock (appetitive → fear) was not significantly different from the response to food after the shock experience (fear → appetitive, *Figure 5—figure supplement 3I*). Similar results were observed for shock, CS^Food, and CS^Shock responses (*Figure 5—figure supplement 3J–L*), suggesting that the encoding of positive or negative stimuli in CeA neurons is not influenced by the previous experience with opposing stimuli.

## Discussion

The CeA is a principally GABAergic structure, allowing us to image the bulk of CeA neurons through conditional expression of GCaMP in Vgat-Cre mice. By imaging a total of 10 mice, we were able to achieve lens placements that largely covered the extent of the dorsal-ventral axis of the CeA. Interestingly, although the CeA is demarcated by functionally distinct subdivisions and genetically distinct cell types, we found that cells responsive to reward and/or fear were intermingled throughout the structure.

The CeA has been proposed to encode both salience and valence (*Fadok et al., 2018*; *Kong and Zweifel, 2021*; *Pignatelli and Beyeler, 2019*), which is supported by studies performing bulk or single cell imaging or recording in the CeA during appetitive or aversive behaviors (*Ciocchi et al., 2010*; *Douglass et al., 2017*; *Duvarci et al., 2011*; *Fadok et al., 2017*; *Hardaway et al., 2019*; *Li et al., 2013*; *Sanford et al., 2017*; *Yang et al., 2023*; *Yu et al., 2017*). The definition of salience or valence encoding is often poorly defined, and the terms are frequently used interchangeably. Moreover, with the exception of a recent study (*Yang et al., 2023*), analysis of specific cell types has not included both an appetitive and aversive stimulus to effectively resolve at the single cell level salience versus valence encoding. For salience encoding, we defined these cells as responding in the same direction to both the appetitive or aversive stimulus as described previously (*Gao et al., 2020*; *Lin and Nicolelis, 2008*; *Steinberg et al., 2020*; *Zhu et al., 2018*). Valence encoding cells were defined as cells responding in either the opposite direction to positive or negative stimuli, or selectively responding to one or the other (*Douglass et al., 2017*; *Isosaka et al., 2015*; *Yu et al., 2017*). Although we observed numerous examples of distinct types of salience and valence encoding cells, the majority of neurons within the CeA were similar to what we previously defined as Type 1 valence neurons (*Kong and Zweifel, 2021*), that is they respond selectively to either the positive or the negative valence of the US.

Consistent with observations reported for learning-dependent changes in somatostatin expressing neurons of the CeA (*Yang et al., 2023*), we observed increased responsiveness in excited neurons and the number of excited neurons of the CeA to the food US on day 10 compared to day 1 of conditioning. Within the last session of Pavlovian appetitive conditioning, we observed enhanced responses in cells inhibited by the food US and CS^Food and diminished excitatory response to the CS^Food. These results may reflect satiety within the session. Indeed, neurons within the CeA have been shown to play important roles in the regulation of feeding cessation (*Cai et al., 2014*). We did not record calcium dynamics in CeA neurons across multiple days of fear conditioning; however, we did observe a within session decrease in the amplitude of fear US excited cells consistent with a desensitization of these neurons (*Yu et al., 2017*).

Numerous studies have identified CeA neurons that respond to the CS following conditioning (*Ciocchi et al., 2010*; *Duvarci et al., 2011*; *Fadok et al., 2017*; *Li et al., 2013*; *Sanford et al., 2017*; *Yu et al., 2017*). We also observed cells that responded to the CS; however, it was evident that these responses occurred in a much smaller number of cells compared to the US responding and with a smaller amplitude. This is consistent with what has been reported previously for specific cell types within the CeA (*Yang et al., 2023*; *Yu et al., 2017*) and does not discount the previously defined importance of plasticity within these cells for fear-related learning (*Li et al., 2013*; *Penzo et al., 2014*; *Sanford et al., 2017*). Our findings do suggest

however that the CeA as a whole is largely tuned to the valence of the US with approximately equal encoding of both positive and negative valences.

# Materials and methods

## Key resources table

| Reagent type (species) or resource | Designation | Source or reference | Identifiers | Additional information |
|---|---|---|---|---|
| Antibody | Anti-GFP (chicken polyclonal) | abcam | RRID:AB_300798 | (1:6000) |
| Antibody | Anti-chicken- AlexaFluor 488 (donkey polyclonal) | Jackson ImmunoResearch | RRID:AB_2340375 | (1:250) |
| Strain, strain background | AAV1-FLEX-GCaMP6m | University of Washington | N/A | |
| Strain, strain background (*Mus musculus*) | B6J.129S6(FVB)-Slc32a1$^{tm2(cre)Lowl}$/MwarJ | Jackson laboratory | RRID:IMSR_JAX:028862 | 4 males and 6 females |
| Software, algorithms | MATLAB | The MathWorks, Inc | RRID:SCR_001622 | |
| Software, algorithms | R | The R Foundation | RRID:SCR_001905 | |
| Software, algorithms | Med-PC | Med Associates, Inc | RRID:SCR_012156 | |
| Software, algorithms | Graph Pad Software | GraphPad Software, Inc | RRID:SCR_002798 | |
| Software, algorithms | Inscopix data processing software | Inscopix | | |
| Software, algorithms | Inscopix data acquisition software | Inscopix | | IDAS 1.5.4 |
| Other | DAPI Fluoromount-G | SouthernBiotech | 0100–20 | DAPI staining |
| Other | Gradient-index (GRIN) lens | Inscopix | 1050–004413 | ProViewTM Integrated Lens 0.6mm x7.3mm |
| Other | Dental cement | Lang Dental | 1530BLK | Contemporary Ortho-Jet Powder BLACK Powder |
| Other | Dental cement | Lang Dental | 1504BLK | Contemporary Ortho-Jet Liquid |
| Other | Anchoring screws | Antrin Miniature Specialties, Inc | AMS 120/1 P-25 | Screws for anchoring a lens |

## Animals

Male and female Vgat$^{IRES-Cre}$ (Slc32a1$^{tm2(cre)Lowl}$/J; JAX strain #: 016962) mice (10 mice; 4 males and 6 females, sample size was determined by 0.05 α, 0.5 effect size and 0.8 power) were group-housed on a 12hr-light/12hr-dark cycle (lights on 7 AM) with ad libitum access to food and water until surgery and behavioral experiment. All experiments were conducted during the light cycle and strictly followed the guidelines set by the University of Washington Animal Care and Use Committee, ensuring ethical treatment of the animals.

## Surgery and virus

Under isoflurane-induced anesthesia, 12–16 week old Vgat$^{IRES-Cre}$ mice at the time of surgery were placed in a stereotaxic instrument (Kopf) and injected with 0.5 μl of AAV1-FLEX-GCaMP6m (produced in-house with a titer of ~3 X $10^{12}$ particles/ml *Gore et al., 2013*). The injection was made in the right CeA (AP: –1.2 mm, ML: 2.9 mm, DV: –4.6 mm) at a rate of 0.25 μl/min. Following virus injection, mice were implanted with a gradient-index lens (GRIN lens with baseplates attached, 0.6 mm diameter, 7.3 length; Inscopix) at the virus injection site. The GRIN lens was fixed with black dental cement (Lang Dental) with anchoring screws (Antrin Miniature Specialties). Four weeks after the surgery, animals were placed on a standard food restriction schedule with free access to water until they reached ~85% of pre-operation weights. Behavioral experiments began 5 weeks after surgery.

## Behavioral paradigms

Throughout the experimental sessions (baseline, appetitive Pavlovian conditioning, Pavlovian fear conditioning, post-learning test; *Figure 1B*), the mice maintained their body weights at ~85% of their pre-operation weights. All sessions were conducted in the same operant chamber (21.6 X 17.8 X 12.7 cm) equipped with a house light, a speaker, a metal bar for fear Pavlovian conditioning, and a food dispenser for appetitive Pavlovian conditioning (MedAssociates). The operant chamber was

placed in a sound-attenuating box, and each session had distinct contextual cues based on its type. Random assignment of mice was done to either the appetitive → fear or fear → appetitive group, indicating the order of valence conditioning.

### Baseline
Prior to any conditioning, we assessed the basal freezing behavior and calcium activity in response to CS^Food and CS^Shock were measured. Two CSs with different frequencies of CSs (either 4 kHz or 12 kHz, 10 s each; counter-balanced) were delivered 10 times, alternating with a 90 s inter-trial interval (ITI). The contextual cues for the baseline session consisted of white walls and floor.

### Pavlovian appetitive conditioning
During this session, CS^Food was paired with a pellet (20 mg sucrose pellet, Bioserv) for 20 trials per day, conducted over a period of 10 days. A 3 s delay was introduced between CS^Food and the presentation of the food pellet to distinguish the offset of CS^Food from the onset of food delivery. The ITI was set to 90 s. The floor of the chamber was covered with a white plastic panel, while the walls were kept transparent. The chamber was cleaned with a chlorine dioxide-based sterilant (Clidox-S) between each animal.

### Pavlovian fear conditioning
In this session, CS^Shock was paired with a foot shock (0.5 mA) for 10 trials, conducted for 1 day. Similar to the appetitive conditioning, a 3 s delay was introduced between CS^Shock and the delivery of the foot shock to distinguish the offset of CS^Shock from the onset of shock. The contextual cues during this session included white walls, a grid floor for shock delivery, and a 1% acetic acid odor cue. The chamber was cleaned with 70% ethanol between each animal.

## Pre-processing calcium imaging data
One week prior to the start of the experiment, imaging parameters such as focal planes and LED power were calibrated by screening calcium activity from GCaMP6m-expressing CeA GABAergic neurons. The screening session involved checking calcium signals using nVoke miniscope (Inscopix) at two focal planes (multiplane imaging) with LED power set at 40–70% depending on the GCaMP6m expression level. Calcium activity was acquired at 10 Hz per plane (20 Hz for alternating acquisition), and this final temporal resolution was used for the imaging data without further down-sampling. The fixed pre-set focal planes and LED power were maintained throughout the entire experiment to ensure consistency.

To prevent prolonged exposure to the LED light, the Med Associates chamber sent a TTL pulse to turn on the LED 30 s before the CS onset and turn it off 30 s after the CS onset. Trial-structured multi-plane imaging data were pre-processed through sequential concatenation and spatial down-sampled (spatial factor 4) using Inscopix Data Processing Software (IDPS). The IDPS was then utilized for the subsequent data processing steps, including motion correction, cell identification, and cell registration between planes or days. The following steps were followed: 1. Motion correction was performed multiple times as necessary until the image stabilized. 2. The motion-corrected TIFF (tag image file format) file was reloaded for easier processing of multiplane registration. 3. Multiplane registration was conducted to eliminate imaging the same cells from multiple planes (minimum spatial correlation = 0.5, temporal correlation = 0.99). 4. Cells were identified with a constrained non-negative matrix factorization algorithm for microendoscopic data (CNMF-E) (*Zhou et al., 2018*). 5. Longitudinal registration was used for day 10 of appetitive and day 1 of aversive learning to compare how the same cell responded to appetitive vs. aversive stimuli.

## Identifying significantly responsive neurons
We wish to identify neurons that are responsive to appetitive or fear stimuli (i.e. that display a difference in activity pre- versus post-stimulus). However, the usual p-values derived from statistical analyses such as t-tests or Wilcoxon rank sum tests assume independence of data points, which is violated for neuronal calcium transients due to an inherent dependency between adjacent transient values. Consequently, a p-value computed assuming independence will lead to highly inflated measures of

significance in the case of dependent data, and thus inaccurate interpretation of the data. A related issue is discussed in the *Harris, 2021* study along with suggested remedies.

Therefore, to quantify the responsiveness of CeA neurons to appetitive vs. aversive stimuli, we rely on an alternative method for generating p-values based upon a variant of 'circular shifting' (*Harris, 2021*). The null hypothesis for this statistical test is that the calcium transients are non-responsive to the stimuli (formally, pre-stimulus and post-stimulus data are drawn from the same distribution). To simulate hypothetical neuron data under the null hypothesis, we generated 'null' calcium activity by circularly shifting the calcium trace of trials from randomly selected neurons (Step 1 in *Figure 2B*, red bar). Thus, the 'null' calcium traces mimic the hypothetical situation where transients in calcium traces are unrelated to the experimental variables, while preserving calcium dependency dynamics.

After generating 'null' calcium activity as described above, we compared the test statistics based upon the observed data to the collection of test statistics obtained under the 'null' calcium activity distribution; test statistics were obtained by summing Wilcoxon rank sum test (WRST) statistics across trials.

We note two substantial differences between our approach and the circular shifting approach of the study (*Harris, 2021*): (i) We generate a null distribution by pooling from the entire population of recorded neurons to fully capture the heterogeneity of possible calcium transients under the null distribution. (ii) Our analysis relies on WRST statistics, which operate on ranks (rather than directly on the calcium transient values).

Details are provided in Algorithm 1. Step 1 formally defines the circular shift operation. Step 2 involves repeatedly applying the circular shift operation to a randomly selected neuron, and then computing the resulting WRST; these results in a distribution of WRSTs generated under the null hypothesis. In Step 3, the WRST test statistics observed in the current study are computed, and in Step 4, they are compared with the null distribution generated in Step 2 to assess their statistical significance.

We now discuss Algorithm 1 in greater detail. In Algorithm 1, N is the number of timepoints for each neuron's transient (note that each neuron has been recorded for the same number of timepoints).

---

**Algorithm 1. Computing a p-value for a neuron's responsiveness**

---

Step 0: Isolate the neural activity of interest.

---

If testing for CS, then consider only neural activity ranging from 30 seconds before CS to 10 seconds after CS.
If testing for food and shock, consider only neural activity ranging from 30 seconds before CS to 30 seconds after CS.

---

Step 1: Define the "circular shift" operation

---

Define CircularShift($Y$, N, s):
$Y^* \leftarrow [Y[(s+1):N], Y[1:s]]$ return($Y^*$)

---

Step 2: Create a repository of test statistics for null data

---

For $b$ in {1, 2, …, B} do
 $i \leftarrow$ random integer from 1 to the number of neurons
 For $j$ in {1,2, …, number of trials} do
 s ← random integer between 1 and N

 $Y^*_{b,j} \leftarrow$ CircularShift($Y_{i,j}$, N, s)

 $W^*_{b,j} \leftarrow$ Wilcoxon rank sum test of a difference in $Y^*_{b,j}$ (pre- vs. post-stimuli)
 end for

 $W^*_b \leftarrow \sum_{X} w^*_{b,j}$

end for

---

Step 3: Calculate observed test statistics

---

For $i$ in {1,2, …, number of neurons} do
 For $j$ in {1,2, …, number of trials} do

 $W^{obs}_{i,j} \leftarrow$ Wilcoxon rank sum test statistic with $Y_{i,j}$ (pre- vs. post-stimuli)
 end for

 $\widetilde{W}^{obs}_i \leftarrow \sum_{j} W^{obs}_{i,j}$

---

*Continued on next page*

*Continued*

---

**Algorithm 1. Computing a p-value for a neuron's responsiveness**

---

Step 4: Compute p-values

---

For i in {1,2,,.., number of neurons} do

$$p_i^+ \leftarrow [(\# \text{ of } W_b^* \geq \widetilde{W}_i^{obs})+1]/[B+1]$$

$$p_i^- \leftarrow [(\# \text{ of } W_b^* \leq \widetilde{W}_i^{obs})+1]/[B+1]$$

$$p_i \leftarrow 2 \times \min(p_i^+, p_i^-)$$

end for

---

(Step 0): We restrict our attention to neural activity in a relevant time window. To test for CS, we consider timepoints ranging from –30 s to 10 s. To test for food and shock, we consider timepoints from –30 s to 30 s. Data outside of these windows is not used in the remaining analysis steps.

(Step 1): We formalize the 'circular shift' operation described earlier.

(Step 2): We repeat the following procedure B=500 times. We randomly select one neuron ($Y_i$) from all the neurons recorded during each session (appetitive: 872, aversive: 518, baseline: 925, post-test: 788), randomly select an integer s, and apply circular shifting by s to each trial for this neuron (this yields $Y_{b,j}^*$, which represents 'null' neuronal activity, as shown in *Figure 2C*).

Next, for the jth trial, we compute the WRST ($W_{b,j}^*$) on the newly-generated 'null' neuronal traces ($Y_{b,j}^*$) to compare pre-stimulus (CSs: –20 s to 0 s; food/shock: –5 s to 13 s) and post-stimulus (CSs: 0 s to 10 s; food: 13s to 30s; shock: 13 s to 18 s) activity (as illustrated in *Figure 2C*). Finally, we sum the WRST statistics across the trials ($\sum_j W_{b,j}^*$) and refer to this as $W_b^*$.

At the end of Step 2, we have $B$ test statistics ($W_1^*, \ldots, W_B^*$), each of which is obtained by summing the WRSTs of a different 'null' neuron across trials. We set B=500 to balance code runtime against having a rich enough null distribution of the test statistic.

(Step 3): For the $j_{th}$ trial of the $i_{th}$ recorded neuron, we compute the WRST between pre- vs. post-stimulus ($W_{i,j}^{obs}$), and then sum these WRST statistics across the trials ($\widetilde{W}_i^{obs}$; *Figure 2D*). The testing window for pre-stimulus and post-stimulus remains the same as in Step 2.

(Step 4): We compare the $i$th neuron's observed test statistic $\left(\widetilde{W}_i^{\text{obs}}\right)$ to the null distribution ($W_b^*$, b=1,...,B) generated in Step 2. The right-tailed p-value $p_i^+$ is calculated as (the number of $W_b^* \geq \widetilde{W}_i^{obs}$ )+1, divided by B+1. Similarly, the left-tailed p-value $p_i^-$ is calculated as (the number of $W_b^* \leq \widetilde{W}_i^{obs}$ )+1, divided by B+1 (*Figure 2E*). We compute the left- and right-tailed p-values separately since $\widetilde{W}_i^{obs}$ can be either extremely large (indicating significant excitation) or small (indicating significant inhibition). For example, if $W_b^* \geq \widetilde{W}_i^{obs}$ for 10 of the 'null' neurons, with B=500, then $p_i^+$ = (10+1) / (500+1)=0.022 and $p_i^-$ = 0.978. The two-sided p-value for this neuron is 2 X 0.022 (since $p_i^+$ is smaller than $p_i^-$), that is it equals 0.044. The significance level is set to 0.05. If the p-value of a neuron is smaller than 0.05, it is considered responsive (significantly excited).

*Figure 5—figure supplement 2* displays the results obtained using the traditional WRST that relies on the theoretical null distribution, that is that assumes independence between time points, and then sets a threshold for the number of trials with a p-value below 0.05 in order to declare that a neuron is 'responsive'. We see little difference between the two sets of results.

## Classification of salience-encoding and valence-encoding neurons

With the neurons that appeared during both appetitive and fear conditioning, we conducted a direct comparison of how individual neurons respond to appetitive vs. aversive stimuli. After the new analysis confirmed 'responsive neurons' (either excited or inhibited) among the registered neurons, we further classified them into salience-encoding and valence-encoding neurons based on their response patterns. Salience-encoding neurons demonstrate significant responses to both appetitive and aversive stimuli in the same direction (both excited or both inhibited), reflecting the stimulus strength, which is the definition of salience (*Figure 5B* white, *Figure 5G* white, *Figure 5—figure supplement 2I and O* white; *Kong and Zweifel, 2021*). On the other hand, valence-encoding neurons exhibit a more diverse range of response types. For instance, some neurons exclusively encode one of the stimuli by

showing a significant response to only one type. Alternatively, some neurons respond significantly to both stimuli but in opposite directions (e.g. excited to food and inhibited to shock, *Figure 5B* black, *Figure 5G* black, *Figure 5—figure supplement 2I and O* black). These response patterns correspond to the definition of valence, representing either positive (good) or negative (bad) valence (*Kong and Zweifel, 2021*). Neurons that showed no significant activity in response to both appetitive and aversive stimuli were categorized as not-significant neurons (*Figure 5B* grey, *Figure 5G* grey, *Figure 5—figure supplement 2I and O* grey).

## Histology

At the conclusion of the experiment, the mice were administered an overdose of Beuthanasia and then perfused transcranially with phosphate-buffered saline (PBS) and 4% paraformaldehyde (PFA). Following this, each mouse head with a GRIN lens was immersed in PFA at 4 °C overnight for a week to preserve the lens track. The brains were subsequently extracted from the post-fixed heads, including the lens track, and cryoprotected in PBS containing 30% sucrose for 72 hr. Immunohistochemistry was performed on transverse sections (50 µm) using a primary antibody (anti-GFP Chicken polyclonal, Abcam, ab13970) followed by a secondary antibody (Alexa Fluor 488, JacksonImmuno, 703-545-155). The treated sections were then mounted on slides and coverslipped with DAPI Fluoromount-G (Southern Biotech). The expression of GCaMP6m and the location of the lens were examined using a Keyence Fluorescence Microscope (Keyence) to determine data inclusion or exclusion animals that lacked GCaMP fluorescence or lens placement in the CeA were excluded.

## Statistical analysis

The responsiveness of CeA neurons to both appetitive and aversive stimuli was quantified using the previously outlined Algorithm 1, implemented in the R programming language. For the determination of statistical significance in behavioral results and area under curve data, we employed One-way repeated measured ANOVA, One-way ANOVA (with Tukey post hoc), paired t-test, and unpaired t-test using Prism software. A comprehensive presentation of the statistical results is available in *Supplementary file 1*. A significance level of $p < 0.05$ was set for all tests. Graphs were generated using GraphPad Prism (version 10) and customized MATLAB codes.

## Acknowledgements

We thank Dr. Selena Schattauer and Dr. James Allen for assistance with viral production. We thank colleagues for thoughtful discussion and insights for the manuscript. This work was supported by funding from the U.S. National Institutes of Health, F32MH127801 (MSK), R01MH104450 (LSZ), and the University of Washington Center for Excellence in Opioid Addiction Research: P30DA048736.

## Additional information

### Funding

| Funder | Grant reference number | Author |
| --- | --- | --- |
| National Institutes of Health | F32MH127801 | Mi-Seon Kong |
| National Institutes of Health | R01MH104450 | Larry S Zweifel |

The funders had no role in study design, data collection and interpretation, or the decision to submit the work for publication.

### Author contributions

Mi-Seon Kong, Conceptualization, Data curation, Formal analysis, Funding acquisition, Validation, Investigation, Visualization, Methodology, Writing – original draft, Writing – review and editing; Ethan

Ancell, Daniela M Witten, Data curation, Formal analysis, Methodology; Larry S Zweifel, Conceptualization, Resources, Supervision, Funding acquisition, Investigation, Methodology, Writing – original draft, Writing – review and editing

### Author ORCIDs
Mi-Seon Kong https://orcid.org/0000-0001-8970-7034
Larry S Zweifel https://orcid.org/0000-0003-3465-5331

### Ethics
All experiments were conducted during the light cycle and strictly followed the guidelines set by the University of Washington Institutional Animal Care and Use Committee (IACUC) under the approved protocol (4249-01) to ensure the ethical treatment of the animals.

Reviewer #2 (Public review): https://doi.org/10.7554/eLife.101980.3.sa1
Reviewer #3 (Public review): https://doi.org/10.7554/eLife.101980.3.sa2
Author response https://doi.org/10.7554/eLife.101980.3.sa3

---

## Additional files

### Supplementary files
MDAR checklist

Supplementary file 1. Results of statistical analyses.

### Data availability
The data supporting the findings of this study are available at Dryad. The circular shifting tools are accessible on GitHub (copy archived at *Zweifel, 2024*) and Dryad.

The following dataset was generated:

| Author(s) | Year | Dataset title | Dataset URL | Database and Identifier |
|---|---|---|---|---|
| Kong M, Ancell E, Witten D, Zweifel LS | 2024 | Valence and salience encoding in the central amygdala | https://doi.org/10.5061/dryad.zgmsbccng | Dryad Digital Repository, 10.5061/dryad.zgmsbccng |

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
