## [Editor Report · eLife Assessment]

This **useful** work reveals differential activity to food and shock outcomes in central amygdala GABAergic neurons. Evidence supports claims of unconditioned stimulus activity that changes with learning. **Compelling** evidence that the circular shift method rigorously identifies functional neuron types is also presented. However, the evidence regarding claims related to valence or salience signaling in these neurons is **incomplete**. This work will be of interest to neuroscientists studying sensory processing and learning in the amygdala.

---

## [Referee Report · Reviewer #2 (Public review)]

This study presents valuable insight on how neurons within the central amygdala may broadly encode the valence of emotional stimuli. The evidence supporting most of the authors' conclusion is solid, although some of the claims should be treated with caution due to potential alternative interpretation of the data.

In this revised manuscript the authors have addressed the reviewers' critiques in a way that acknowledges the feedback but does not fully embrace or rigorously address the reviewers' core concerns. Here are the main observations that support this impression:

(1) The authors repeatedly acknowledge the ambiguity in defining "valence" and "salience" in the literature, but their responses don't clarify how they address these terms more rigorously. They seem to justify their operational definitions by citing previous studies but do not address how their definitions impact the clarity and robustness of their findings.

(2) The reviewers highlighted that using stimuli from different sensory modalities without scaling them or including neutral cues limits the ability to distinguish between valence and salience. The authors acknowledge this but argue that using same-modality stimuli would not produce distinct responses. This response doesn't address the reviewers' point about how these design limitations could weaken the conclusions. They seem to rely on citations of similar experimental designs instead of addressing the core critique or proposing additional experiments.

(3) In response to the low number of cue-responsive units and the call for more rigorous behavioral measures (like licking or orienting), the authors provide some data but emphasize statistical rigor over behavioral insights, which was questioned during the initial review. They don't propose any methodological adjustments or consider alternative explanations.

(4) The reviewers suggested clustering or other population-level analyses to understand functional diversity within the central amygdala. The authors argue that their statistical approach was sufficient and don't believe additional clustering analyses would add value. This response seems dismissive, as they don't consider whether population-level insights might reveal patterns that single-cell responses overlook.

Overall, while the authors have responded to each concern, their rebuttals often reference other studies to justify their choices rather than addressing the specific limitations highlighted by the reviewers.

---

## [Referee Report · Reviewer #3 (Public review)]

Summary:

The authors have performed endoscopic calcium recordings of individual CeA neuron responses to food and shock, as well as to cues predicting food and shock. They claim that a majority of neurons encode valence, with a substantial minority encoding salience.

Strengths:

The use of endoscopic imaging is valuable, as it provides the ability to resolve signals from single cells, while also being able to track these cells across time (though the latter capability was not extensively utilized). Another strength is the use of a sophisticated circular shifting analysis to avoid statistical errors caused by correlations between neighboring image pixels.

Weaknesses:

In the first version of this manuscript, my main critique was that the authors didn't fully test whether neurons encode valence. In their rebuttal, the authors justify their use of the terms valence and salience by citing prior works from different labs:

(1) Li et al., 2019, doi: 10.7554/eLife.41223

(2) Yang et al., 2023, doi: 10.1038/s41586-023-05910-2

(3) Huang et al., 2024, doi: 10.1038/s41586-024-07819

(4) Lin and Nicolelis, 2008, doi: 10.1016/j.neuron.2008.04.031

(5) Stephenson-Jones et al., 2020, doi: 10.1016/j.neuron.2019.12.006

(6) Zhu et al., 2018, doi: 10.1126/science.aat0481

(7) Comoli et al., 2003, doi: 10.1038/nn1113P

Among these, items #1 and #3 primarily discuss valence, while #2, #4, #6, and #7 discuss salience, and #5 discusses both.

Upon reviewing these references, the authors' identification of valence encoding patterns is still problematic, and indeed studies cited above show several lines of evidence for valence encoding that are absent here. For example, item #3 ranked behavioral responses to five different odors in *Drosophila*, from most attractive to most repulsive, and saw neuronal responses correlated with the degree of attraction versus repulsion across all five odors. This is robust evidence for valence encoding that is absent here. Items #1 and #5 above are the other two valence-addressing studies cited, and although those only used one rewarding and one aversive stimulus (in rodents), both also added a neutral cue, and most critically, identified substantial subsets of neurons showing a rank-order response, e.g. either aversion > neutral > reward or aversion < neutral < reward. Again, that level of demonstration of valence encoding is not shown in the current study.

Finally, two of the valence studies above tested responses to omission of reward/punishment, providing yet more evidence of valence encoding that is absent in the current study.

While there is much to like about the current study, the claims of valence encoding appear hard to justify, and should be toned down.

---

## [Author Response]

The following is the authors’ response to the original reviews.

**Reviewer #1 (Public review):**

**From the Reviewing Editor:**
Four reviewers have assessed your manuscript on valence and salience signaling in the central amygdala. There was universal agreement that the question being asked by the experiment is important. There was consensus that the neural population being examined (GABA neurons) was important and the circular shift method for identifying task-responsive neurons was rigorous. Indeed, observing valenced outcome signaling in GABA neurons would considerably increase the role the central amygdala in valence. However, each reviewer brought up significant concerns about the design, analysis and interpretation of the results. Overall, these concerns limit the conclusions that can be drawn from the results. Addressing the concerns (described below) would work towards better answering the question at the outset of the experiment: how does the central amygdala represent salience vs valence.A weakness noted by all reviewers was the use of the terms 'valence' and 'salience' as well as the experimental design used to reveal these signals. The two outcomes used emphasized non-overlapping sensory modalities and produced unrelated behavioral responses. Within each modality there are no manipulations that would scale either the value of the valenced outcomes or the intensity of the salient outcomes. While the food outcomes were presented many times (20 times per session over 10 sessions of appetitive conditioning) the shock outcomes were presented many fewer times (10 times in a single session). The large difference in presentations is likely to further distinguish the two outcomes. Collectively, these experimental design decisions meant that any observed differences in central amygdala GABA neuron responding are unlikely to reflect valence, but likely to reflect one or more of the above features.

We appreciate the reviewers’ comments regarding the experimental design. When assessing fear versus reward, we chose stimuli that elicit known behavioral responses, freezing versus consumption. The use of stimuli of the same modality is unlikely to elicit easily definable fear or reward responses or to be precisely matched for sensory intensity. For example, sweet or bitter tastes can be used, but even these activate different taste receptors and vary in the duration of the activation of taste-specific signaling (e.g. how long the taste lingers in the mouth). The approach we employed is similar to that of Yang et al., 2023 (doi: 10.1038/s41586-023-05910-2) that used water reward and shock to characterize the response profiles of somatostatin neurons of the central amygdala. Similar to what was reported by Yang and colleagues we observed that the majority of CeA GABA neurons responded selectively to one unconditioned stimulus (~52%). We observed that 15% of neurons responded in the same direction, either activated or inhibited, by the food or shock US. These were defined as salience based on the definitions of Lin and Nicolelis, 2008 (doi: 10.1016/j.neuron.2008.04.031) in which basal forebrain neurons responded similarly to reward or punishment irrespective of valence. The designation of valence encoding based opposite responses to the food or shock is straightforward (~10% of cells); however, we agree that the designation of modality-specific encoding neurons as valence encoding is less straightforward.

A second weakness noted by a majority of reviewers was a lack of cue-responsive unit and a lack of exploration of the diversity of response types, and the relationship cue and outcome firing. The lack of large numbers of neurons increasing firing to one or both cues is particularly surprising given the critical contribution of central amygdala GABA neurons to the acquisition of conditioned fear (which the authors measured) as well as to conditioned orienting (which the authors did not measure). Regression-like analyses would be a straightforward means of identifying neurons varying their firing in accordance with these or other behaviors. It was also noted that appetitive behavior was not measured in a rigorous way. Instead of measuring time near hopper, measures of licking would have been better. Further, measures of orienting behaviors such as startle were missing.The authors also missed an opportunity for clustering-like analyses which could have been used to reveal neurons uniquely signaling cues, outcomes or combinations of cues and outcomes. If the authors calcium imaging approach is not able to detect expected central amygdala cue responding, might it be missing other critical aspects of responding?

As stated in the manuscript, we were surprised by the relatively low number of cue responsive cells; however, when using a less stringent statistical method (Figure 5 - Supplement 2), we observed 13% of neurons responded to the food associated cue and 23% responded to the shock associated cue. The differences are therefore likely a reflection of the rigor of the statistical measure to define the responsive units. The number of CS responsive units is less than reported in the CeAl by Ciocchi et al., 2010 (doi: 10.1038/nature09559) who observed 30% activated by the CS and 25% inhibited, but is not that dissimilar from the results of Duvarci et al., 2011 (doi: 10.1523/JNEUROSCI.4985-10.2011) who observed 11% activated in the CeAl and 25% inhibited by the CS. These numbers are also consistent with previous single cell calcium imaging of cell types in the CeA. For example, Yang et al., 2023 (doi: 10.1038/s41586-023-05910-2) observed that 13% of somatostatin neurons responded to a reward CS and 8% responded to a shock CS. Yu et al., 2017 (doi: 10.1038/s41593-017-0009-9) observed 26.5% of PKCdelta neurons responded to the shock CS. It should also be noted that our analysis was not restricted to the CeAl. Finally, Food learning was assessed in an operant chamber in freely moving mice with reward pellet delivery. Because liquids were not used for the reward US, licking is not a metric that can be used.

All reviewers point out that the evidence for salience encoding is even more limited than the evidence for valence. Although the specific concern for each reviewer varied, they all centered on an oversimplistic definition of salience. Salience ought to scale with the absolute value and intensity of the stimulus. Salience cannot simply be responding in the same direction. Further, even though the authors observed subsets of central amygdala neurons increasing or decreasing activity to both outcomes - the outcomes can readily be distinguished based on the temporal profile of responding.

We thank the reviewers for their comments relating to the definition of salience and valence encoding by central amygdala neurons. We have addressed each of the concerns below.

Additional concerns are raised by each reviewer. Our consensus is that this study sought to answer an important question - whether central amygdala signal salience or valence in cue-outcome learning. However, the experimental design, analyses, and interpretations do not permit a rigorous and definitive answer to that question. Such an answer would require additional experiments whose designs would address the significant concerns described here. Fully addressing the concerns of each reviewer would result in a re-evaluation of the findings. For example, experimental design better revealing valence and salience, and analyses describing diversity of neuronal responding and relationship to behavior would likely make the results Important or even Fundamental.

We appreciate the reviewers’ comments and have addressed each concern below.

**Reviewer #2 (Public review):**
In this article, Kong and authors sought to determine the encoding properties of central amygdala (CeA) neurons in response to oppositely valenced stimuli and cues predicting those stimuli. The amygdala and its subregional components have historically been understood to be regions that encode associative information, including valence stimuli. The authors performed calcium imaging of GABA-ergic CeA neurons in freely-moving mice conditioned in Pavlovian appetitive and fear paradigms, and showed that CeA neurons are responsive to both appetitive and aversive unconditioned and conditioned stimuli. They used a variant of a previously published 'circular shifting' technique (Harris, 2021), which allowed them to delineate between excited/non-responsive/inhibited neurons. While there is considerable overlap of CeA neurons responding to both unconditioned stimuli (in this case, food and shock, deemed "salience-encoding" neurons), there are considerably fewer CeA neurons that respond to both conditioned stimuli that predict the food and shock. The authors finally demonstrated that there are no differences in the order of Pavlovian paradigms (fear - shock vs. shock - fear), which is an interesting result, and convincingly presented given their counterbalanced experimental design.In total, I find the presented study useful in understanding the dynamics of CeA neurons during a Pavlovian learning paradigm. There are many strengths of this study, including the important question and clear presentation, the circular shifting analysis was convincing to me, and the manuscript was well written. We hope the authors will find our comments constructive if they choose to revise their manuscript.While the experiments and data are of value, I do not agree with the authors interpretation of their data, and take issue with the way they used the terms "salience" and "valence" (and would encourage them to check out Namburi et al., NPP, 2016) regarding the operational definitions of salience and valence which differ from my reading of the literature. To be fair, a recent study from another group that reports experiments/findings which are very similar to the ones in the present study (Yang et al., 2023, describing valence coding in the CeA using a similar approach) also uses the terms valence and salience in a rather liberal way that I would also have issues with (see below). Either new experiments or revised claims would be needed here, and more balanced discussion on this topic would be nice to see, and I felt that there were some aspects of novelty in this study that could be better highlighted (see below).One noteworthy point of alarm is that it seems as if two data panels including heatmaps are duplicated (perhaps that panel G of Figure 5-figure supplement 2 is a cut and paste error? It is duplicated from panel E and does not match the associated histogram).

We thank the reviewer for their insightful comments and assessment of the manuscript.

Major concerns:(1) The authors wish to make claims about salience and valence. This is my biggest gripe, so I will start here.(1a) Valence scales for positive and negative stimuli and as stated in Namburi et al., NPP, 2016 where we operationalize "valence" as having different responses for positive and negative values and no response for stimuli that are not motivational significant (neutral cues that do not predict an outcome). The threshold for claiming salience, which we define as scaling with the absolute value of the stimulus, and not responding to a neutral stimulus (Namburi et al., NPP, 2016; Tye, Neuron, 2018; Li et al., Nature, 2022) would require the lack of response to a neutral cue.

We appreciate the reviewer’s comment on the definitions of salience and valence and agree that there is not a consistent classification of these response types in the field. As stated above, we used the designation of salience encoding if the cells respond in the same direction to different stimuli regardless of the valence of the stimulus similar to what was described previously (Lin and Nicolelis, 2008, doi: 10.1016/j.neuron.2008.04.031). Similar definitions of salience have also been reported elsewhere (for examples see: Stephenson-Jones et al., 2020, doi: 10.1016/j.neuron.2019.12.006, Zhu et al., 2018 doi: 10.1126/science.aat0481, and Comoli et al., 2003, doi: 10.1038/nn1113P). Per the suggestion of the reviewer, we longitudinally tracked cells on the first day of Pavlovian reward conditioning the fear conditioning day. Although there were considerably fewer head entries on the first day of reward conditioning, we were able to identify 10 cells that were activated by both the food US and shock US. We compared the responses to the first five head entries and last head entries and the first 5 shocks and last five shocks. Consistent with what has been reported for salience encoding neurons in the basal forebrain (Lin and Nicolelis, 2008, doi: 10.1016/j.neuron.2008.04.031), we observed that the responses were highest when the US was most unexpected and decreased in later trials.

(1b) The other major issue is that the authors choose to make claims about the neural responses to the USs rather than the CSs. However, being shocked and receiving sucrose also would have very different sensorimotor representations, and any differences in responses could be attributed to those confounds rather than valence or salience. They could make claims regarding salience or valence with respect to the differences in the CSs but they should restrict analysis to the period prior to the US delivery.

Perhaps the reviewer missed this, but analysis of valence and salience encoding to the different CSs are presented in Figure 5G, Figure 5 -Supplement 1 C-D, and Figure 5 -Supplement 2 N-O. Analysis of CS responsiveness to CSFood and CSShock were analyzed during the conditioning sessions Figure 3E-F, Figure 4B-C, Figure 5 – Supplement 2J-O and Figure 5 – Supplement 3K-L, and during recall probe tests for both CSFood and CSShock, Figure 5 – Supplement 1C-J.

(1c) The third obstacle to using the terms "salience" or "valence" is the lack of scaling, which is perhaps a bigger ask. At minimum either the scaling or the neutral cue would be needed to make claims about valence or salience encoding. Perhaps the authors disagree - that is fine. But they should at least acknowledge that there is literature that would say otherwise.(1d) In order to make claims about valence, the authors must take into account the sensory confound of the modality of the US (also mentioned in Namburi et al., 2016). The claim that these CeA neurons are indeed valence-encoding (based on their responses to the unconditioned stimuli) is confounded by the fact that the appetitive US (food) is a gustatory stimulus while the aversive US (shock) is a tactile stimulus.

We provided the same analysis for the US and CS. The US responses were larger and more prevalent, but similar types of encoding were observed for the CS. We agree that the food reward and the shock are very different sensory modalities. As stated above, the use of stimuli of the same modality is unlikely to elicit easily definable fear or reward responses or to be precisely matched for sensory intensity. We agree that the definition of cells that respond to only one stimulus is difficult to define in terms of valence encoding, as opposed to being specific for the sensory modality and without scaling of the stimulus it is difficult to fully address this issue. It should be noted however, that if the cells in the CeA were exclusively tuned to stimuli of different sensory modalities, we would expect to see a similar number of cells responding to the CS tones (auditory) as respond to the food (taste) and shock (somatosensory) but we do not. Of the cells tracked longitudinally 80% responded to the USs, with 65% of cells responding to food (activated or inhibited) and 44% responding to shock (activated or inhibited).

(2) Much of the central findings in this manuscript have been previously described in the literature. Yang et al., 2023 for instance shows that the CeA encodes salience (as demonstrated by the scaled responses to the increased value of unconditioned stimuli, Figure 1 j-m), and that learning amplifies responsiveness to unconditioned stimuli (Figure 2). It is nice to see a reproduction of the finding that learning amplifies CeA responses, though one study is in SST::Cre and this one in VGAT::cre - perhaps highlighting this difference could maximize the collective utility for the scientific community?

We agree that the analysis performed here is similar to what was conducted by Yang et al., 2023. With the major difference being the types of neurons sampled. Yang et al., imaged only somatostatin neurons were as we recorded all GABAergic cell types within the CeA. Moreover, because we imaged from 10 mice, we sampled neurons that ostensibly covered the entire dorsal to ventral extent of the CeA (Figure 1 – Supplement 1). Remarkably, we found that the vast majority of CeA neurons (80%) are responsive to food or shock. Within this 80% there are 8 distinct response profiles consistent with the heterogeneity of cell types within the CeA based on connectivity, electrophysiological properties, and gene expression. Moreover, we did not find any spatial distinction between food or shock responsive cells, with the responsive cell types being intermingled throughout the dorsal to ventral axis (Figure 5 – Supplement 3).

(3) There is at least one instance of copy-paste error in the figures that raised alarm. In the supplementary information (Figure 5- figure supplement 2 E;G), the heat maps for food-responsive neurons and shock-responsive neurons are identical. While this almost certainly is a clerical error, the authors would benefit from carefully reviewing each figure to ensure that no data is incorrectly duplicated.

We thank the reviewer for catching this error. It has been corrected.

(4) The authors describe experiments to compare shock and reward learning; however, there are temporal differences in what they compare in Figure 5. The authors compare the 10th day of reward learning with the 1st day of fear conditioning, which effectively represent different points of learning and retrieval. At the end of reward conditioning, animals are utilizing a learned association to the cue, which demonstrates retrieval. On the day of fear conditioning, animals are still learning the cue at the beginning of the session, but they are not necessarily retrieving an association to a learned cue. The authors would benefit from recording at a later timepoint (to be consistent with reward learning- 10 days after fear conditioning), to more accurately compare these two timepoints. Or perhaps, it might be easier to just make the comparison between Day 1 of reward learning and Day 1 of fear learning, since they must already have these data.

We agree that there are temporal differences between the food and shock US deliveries. This is likely a reflection of the fact that the shock delivery is passive and easily resolved based on the time of the US delivery, whereas the food responses are variable because they are dependent upon the consumption of the sucrose pellet. Because of these differences the kinetics of the responses cannot be accurately compared. This is why we restricted our analysis to whether the cells were food or shock responsive. Aside from reporting the temporal differences in the signals did not draw major conclusions about the differences in kinetics. In our experimental design we counterbalanced the animals that received fear conditioning firs then food conditioning, or food conditioning then fear conditioning to ensure that order effects did not influence the outcome of the study. It is widely known that Pavlovian fear conditioning can facilitate the acquisition of conditioned stimulus responses with just a single day of conditioning. In contrast, Pavlovian reward conditioning generally progresses more slowly. Because of this we restricted our analysis to the last day of reward conditioning to the first and only day of fear conditioning. However, as stated above, we compared the responses of neurons defined as salience during day 1 of reward conditioning and fear conditioning. As would be predicted based on previous definitions of salience encoding (Lin and Nicolelis, 2008, doi: 10.1016/j.neuron.2008.04.031), we observed that the responses were highest when the US was most unexpected.

(5) The authors make a claim of valence encoding in their title and throughout the paper, which is not possible to make given their experimental design. However, they would greatly benefit from actually using a decoder to demonstrate their encoding claim (decoding performance for shock-food versus shuffled labels) and simply make claims about decoding food-predictive cues and shock-predictive cues. Interestingly, it seems like relatively few CeA neurons actually show differential responses to the food and shock CSs, and that is interesting in itself.

As stated above, valence and salience encoding were defined similar to what has been previously reported (Li et al., 2019, doi: 10.7554/eLife.41223; Yang et al., 2023, doi: 10.1038/s41586-023-05910-2; Huang et al., 2024, doi: 10.1038/s41586-024-07819; Lin and Nicolelis, 2008, doi: 10.1016/j.neuron.2008.04.031; Stephenson-Jones et al., 2020, doi: 10.1016/j.neuron.2019.12.006; Zhu et al., 2018, doi: 10.1126/science.aat0481; and Comoli et al., 2003, doi: 10.1038/nn1113P). Interestingly, many of these studies did not vary the US intensity.

**Reviewer #3 (Public review):**
Summary:In their manuscript entitled Kong and colleagues investigate the role of distinct populations of neurons in the central amygdala (CeA) in encoding valence and salience during both appetitive and aversive conditioning. The study expands on the work of Yang et al. (2023), which specifically focused on somatostatin (SST) neurons of the CeA. Thus, this study broadens the scope to other neuronal subtypes, demonstrating that CeA neurons in general are predominantly tuned to valence representations rather than salience.

We thank the reviewer for their insightful comments and assessment of the manuscript.

Strengths:One of the key strengths of the study is its rigorous quantitative approach based on the "circular-shift method", which carefully assesses correlations between neural activity and behavior-related variables. The authors' findings that neuronal responses to the unconditioned stimulus (US) change with learning are consistent with previous studies (Yang et al., 2023). They also show that the encoding of positive and negative valence is not influenced by prior training order, indicating that prior experience does not affect how these neurons process valence.Weaknesses:However, there are limitations to the analysis, including the lack of population-based analyses, such as clustering approaches. The authors do not employ hierarchical clustering or other methods to extract meaning from the diversity of neuronal responses they recorded. Clustering-based approaches could provide deeper insights into how different subpopulations of neurons contribute to emotional processing. Without these methods, the study may miss patterns of functional specialization within the neuronal populations that could be crucial for understanding how valence and salience are encoded at the population level.

We appreciate the reviewer’s comments regarding clustering-based approaches. In order to classify cells as responsive to the US or CS we chose to develop a statistically rigorous method for classifying cell response types. Using this approach, we were able to define cell responses to the US and CS. Importantly, we identified 8 distinct response types to the USs. It is not clear how additional clustering analysis would improve cell classifications.

Furthermore, while salience encoding is inferred based on responses to stimuli of opposite valence, the study does not test whether these neuronal responses scale with stimulus intensity-a hallmark of classical salience encoding. This limits the conclusions that can be drawn about salience encoding specifically.

As stated above, we used salience classifications similar to those previously described (Lin and Nicolelis, 2008, doi: 10.1016/j.neuron.2008.04.031; Stephenson-Jones et al., 2020, doi: 10.1016/j.neuron.2019.12.006; Zhu et al., 2018, doi: 10.1126/science.aat0481; and Comoli et al., 2003, doi: 10.1038/nn1113P). We agree that varying the stimulus intensity would provide a more rigorous assessment of salience encoding; however, several of the studies mentioned above classify cells as salience encoding without varying stimulus intensity. Additionally, the inclusion of recordings with varying US intensities on top of the Pavlovian reward and fear conditioning would further decrease the number of cells that can be longitudinally tracked and would likely decrease the number of cells that could be classified.

In sum, while the study makes valuable contributions to our understanding of CeA function, the lack of clustering-based population analyses and the absence of intensity scaling in the assessment of salience encoding are notable limitations.
**Reviewer #4 (Public review):**
Summary:The authors have performed endoscopic calcium recordings of individual CeA neuron responses to food and shock, as well as to cues predicting food and shock. They claim that a majority of neurons encode valence, with a substantial minority encoding salience.Strengths:The use of endoscopic imaging is valuable, as it provides the ability to resolve signals from single cells, while also being able to track these cells across time. The recordings appear well-executed, and employ a sophisticated circular shifting analysis to avoid statistical errors caused by correlations between neighboring image pixels.Weaknesses:My main critique is that the authors didn't fully test whether neurons encode valence. While it is true that they found CeA neurons responding to stimuli that have positive or negative value, this by itself doesn't indicate that valence is the primary driver of neural activity. For example, they report that a majority of CeA neurons respond selectively to either the positive or negative US, and that this is evidence for "type I" valence encoding. However, it could also be the case that these neurons simply discriminate between motivationally relevant stimuli in a manner unrelated to valence per se. A simple test of this would be to check if neural responses generalize across more than one type of appetitive or aversive stimulus, but this was not done. The closest the authors came was to note that a small number of neurons respond to CS cues, of which some respond to the corresponding US in the same direction. This is relegated to the supplemental figures (3 and 4), and it is not noted whether the the same-direction CS-US neurons are also valence-encoding with respect to different USs. For example, are the neurons excited by CS-food and US-food also inhibited by shock? If so, that would go a long way toward classifying at least a few neurons as truly encoding valence in a generalizable way.

As stated above, valence and salience encoding were defined similar to what has been previously reported (Li et al., 2019, doi: 10.7554/eLife.41223; Yang et al., 2023, doi: 10.1038/s41586-023-05910-2; Huang et al., 2024, doi: 10.1038/s41586-024-07819; Lin and Nicolelis, 2008, doi: 10.1016/j.neuron.2008.04.031; Stephenson-Jones et al., 2020, doi: 10.1016/j.neuron.2019.12.006; Zhu et al., 2018, doi: 10.1126/science.aat0481; and Comoli et al., 2003, doi: 10.1038/nn1113P). As reported in Figure 5 and Figure 5 – Supplement 3, ~29% of CeA neurons responded to both food and shock USs (15% in the same direction and 13.5% in the opposite direction). In contrast, only 6 of 303 cells responded to both the CSfood and CSshock, all in the same direction.

A second and related critique is that, although the authors correctly point out that definitions of salience and valence are sometimes confused in the existing literature, they then go on themselves to use the terms very loosely. For example, the authors define these terms in such a way that every neuron that responds to at least one stimulus is either salience or valence-encoding. This seems far too broad, as it makes essentially unfalsifiable their assertion that the CeA encodes some mixture of salience and valence. I already noted above that simply having different responses to food and shock does not qualify as valence-encoding. It also seems to me that having same-direction responses to these two stimuli similarly does not quality a neuron as encoding salience. Many authors define salience as being related to the ability of a stimulus to attract attention (which is itself a complex topic). However, the current paper does not acknowledge whether they are using this, or any other definition of salience, nor is this explicitly tested, e.g. by comparing neural response magnitudes to any measure of attention.

As stated in response to reviewer 2, we longitudinally tracked cells on the first day of Pavlovian reward conditioning the fear conditioning day. Although there were considerably fewer head entries on the first day of reward conditioning, we were able to identify 10 cells that were activated by both the food US and shock US. We compared the responses to the first five head entries and last head entries and the first 5 shocks and last five shocks. Consistent with what has been reported for salience encoding neurons in the basal forebrain (Lin and Nicolelis, 2008, doi: 10.1016/j.neuron.2008.04.031), we observed that the responses were highest when the US was most unexpected and decreased in later trials.

The impression I get from the authors' data is that CeA neurons respond to motivationally relevant stimuli, but in a way that is possibly more complex than what the authors currently imply. At the same time, they appear to have collected a large and high-quality dataset that could profitably be made available for additional analyses by themselves and/or others.Lastly, the use of 10 daily sessions of training with 20 trials each seems rather low to me. In our hands, Pavlovian training in mice requires considerably more trials in order to effectively elicit responses to the CS. I wonder if the relatively sparse training might explain the relative lack of CS responses?

It is possible that learning would have occurred more quickly if we had used greater than 20 trials per session. However, we routinely used 20-25 trials for Pavlovian reward conditioning (doi: 10.1073/pnas.1007827107; doi: 10.1523/JNEUROSCI.5532-12.2013; doi: 10.1016/j.neuron.2013.07.044; and doi: 10.1016/j.neuron.2019.11.024).